# Anti-Idiotypic VHHs and VHH-CAR-T Cells to Tackle Multiple Myeloma: Different Applications Call for Different Antigen-Binding Moieties

**DOI:** 10.3390/ijms25115634

**Published:** 2024-05-22

**Authors:** Heleen Hanssens, Fien Meeus, Emma L. Gesquiere, Janik Puttemans, Yannick De Vlaeminck, Kim De Veirman, Karine Breckpot, Nick Devoogdt

**Affiliations:** 1Molecular Imaging and Therapy Research Group (MITH), Department of Biomedical Sciences, Vrije Universiteit Brussel, Laarbeeklaan 103/K0, 1090 Brussels, Belgium; heleen.hanssens@vub.be (H.H.); emma.lisette.gesquiere@vub.be (E.L.G.); janik.puttemans@vub.be (J.P.); 2Laboratory for Molecular and Cellular Therapy (LMCT), Translational Oncology Research Center, Department of Biomedical Sciences, Vrije Universiteit Brussel, Laarbeeklaan 103/E2, 1090 Brussels, Belgium; fien.meeus@vub.be (F.M.); yannick.de.vlaeminck@vub.be (Y.D.V.); karine.breckpot@vub.be (K.B.); 3Laboratory for Hematology and Immunology (HEIM), Translational Oncology Research Center, Department of Biomedical Sciences, Vrije Universiteit Brussel, Laarbeeklaan 103/D0, 1090 Brussels, Belgium; kim.de.veirman@vub.be

**Keywords:** multiple myeloma, chimeric antigen receptor, adoptive cell therapy, hematology, CAR-T cells, immuno-oncology, idiotype, VHH

## Abstract

CAR-T cell therapy is at the forefront of next-generation multiple myeloma (MM) management, with two B-cell maturation antigen (BCMA)-targeted products recently approved. However, these products are incapable of breaking the infamous pattern of patient relapse. Two contributing factors are the use of BCMA as a target molecule and the artificial scFv format that is responsible for antigen recognition. Tackling both points of improvement in the present study, we used previously characterized VHHs that specifically target the idiotype of murine 5T33 MM cells. This idiotype represents one of the most promising yet challenging MM target antigens, as it is highly cancer- but also patient-specific. These VHHs were incorporated into VHH-based CAR modules, the format of which has advantages compared to scFv-based CARs. This allowed a side-by-side comparison of the influence of the targeting domain on T cell activation. Surprisingly, VHHs previously selected as lead compounds for targeted MM radiotherapy are not the best (CAR-) T cell activators. Moreover, the majority of the evaluated VHHs are incapable of inducing any T cell activation. As such, we highlight the importance of specific VHH selection, depending on its intended use, and thereby raise an important shortcoming of current common CAR development approaches.

## 1. Introduction

Multiple myeloma (MM) is a hematological malignancy of final-stage matured B cells, also termed plasma cells, that primarily reside in the bone marrow [1,2]. In healthy individuals, plasma cells are responsible for antibody production. After malignant transformation, these cells produce an excessive amount of a dysfunctional antibody, termed M-protein or paraprotein, which is released into the bloodstream and is, in turn, largely responsible for the renal dysfunction observed in patients [3,4]. Further symptoms include anemia, hypercalcemia, and bone lesions [4,5,6,7]. MM accounts for around 10% of all hematological cancers and ranks as the second most diagnosed one [2,8]. Currently, much effort is being made in developing novel therapies, yet MM is still considered incurable, as patients typically relapse repeatedly with increased tolerance to administered drugs.

Current myeloma treatment consists of induction therapy, which mostly comprises a combination of a proteasome inhibitor (PI) (e.g., bortezomib), a corticosteroid (e.g., dexamethasone), and an immunomodulatory drug (IMID) (e.g., thalidomide), most preferably followed by autologous stem cell transplantation (ASCT) [1,2,9]. Further treatment consists of complex combinations of PIs, IMIDs, and (cortico)steroids, potentially supplemented with monoclonal antibodies (mAbs), i.e., daratumumab targeting CD38 or elotuzumab targeting SLAMF7/CS1 [2]. More recently, thriving on the successes in other hematologic cancers (namely, lymphomas and leukemias), important progress has been made in the field of chimeric antigen receptor (CAR)-T cell therapy. In 2021 and 2022, two CAR-T cell products were FDA- and EMA-approved for MM after at least three prior lines of treatment [10]. Both Abecma^®^ (idecabtagene vicleucel) and Carvykti^®^ (ciltacabtagene autoleucel) are second-generation CAR-T cells directed against BCMA and show impressive results in heavily pre-treated patients. However, the success of these novel therapy approaches is, again, limited in time, and patient relapse is still frequently reported [10,11,12]. The reasons for relapse are multifold. Firstly, BCMA-specific CAR-T cell therapy relapse is observed when an anti-CAR immune response occurs. In many cases, including in MM, anti-murine single-chain variable fragment (scFv) antibodies are reported to be responsible for neutralizing the therapeutic cells [10,11,13]. Furthermore, the intrinsically unstable nature of the scFv format renders these CARs susceptible to antigen-independent membrane aggregation [14,15,16]. This may lead to so-called “tonic signaling”, when the T cells go into a chronic but nonspecific state of activation. The result is the early exhaustion of the T cells, which manifests in a reduced cytotoxic effector T cell phenotype and the upregulation of immunosuppressive receptors such as PD1, CTLA4, and LAG3. Premature T cell exhaustion has been mentioned as the second major reason for patient relapse [11,17]. These two findings highlight the notion that the classical scFv-based CAR format may not be ideal [18]. Thirdly, the use of BCMA as a drug target has also been questioned [10,11,12], as it is sensitive to γ-secretase-mediated shedding [19,20,21], and (in fewer cases) antigen loss has also been observed [11,22]. Finally, patient relapse may also be linked to an enhanced immunosuppressive bone marrow micro-environment [11].

A typical CAR consists of an scFv for antigen recognition, a hinge region, a transmembrane domain, and intracellular co-stimulation and T cell activation units [18]. The above-mentioned limitations of using scFvs in CARs (i.e., concerns about immunogenicity and stability) are becoming increasingly recognized, and alternatives are rapidly emerging, as reviewed elsewhere [18]. In particular, the use of camelid heavy chain-only antibody-derived variable heavy domains of heavy chain (VHHs) instead of scFvs is promising. Because these are mAb-derived fragments, their mode of action does not differ tremendously from already clinically successful CAR-T cell products. Furthermore, they evade artificial formatting, which is required for scFv generation, as they are monomeric by nature. In addition to not requiring the inclusion of an artificial linker, VHH sequences’ similarity to human V_H_-III family domains further reduces the likelihood of anti-VHH-CAR immune responses [23,24,25]. 

BCMA is expressed by late-stage B cells and long-lived plasma cells and is upregulated in MM cells, where it is known to have a role in cell survival and proliferation [26]. However, enhanced membrane shedding and genomic shut-down of BCMA expression have been reported as causes of CAR-T cell therapy relapse [19,27]. Therefore, alternative MM (CAR) target antigens are being investigated. Ideally, the targeted antigen is expressed exclusively and unconditionally by the cancer cells. Since plasma cells (and, therefore, also MM cells) are terminally differentiated B cells, they express a terminally differentiated B-cell receptor (BCR) [28]. The BCR possesses a protein stretch that is unique to the receptor and, thus, also to the B cell. In healthy B cells, this protein stretch is responsible for making binding contacts with the epitope on the target protein. It is made up of three hypervariable complementary determining regions (CDRs) on both the variable heavy (V_H_) and variable light (V_L_) domains of the BCR. This combined antigen-recognizing region (i.e., made up of six CDRs in total) is also called the BCR idiotype [29]. Each MM clone has its own idiotype, which is highly interesting, as its expression is cancer cell-exclusive [30]. 

As previously stated, the BCR is shed in excessive amounts from the B cell surface during disease (i.e., as paraprotein or M-protein), even acting as a diagnostic measure for MM. Nonetheless, in a significant portion of patients, a membrane-anchored form of the M-protein is also expressed by MM cells in the bone marrow, including by dormant MM cells [31,32,33,34,35]. An obvious hurdle for idiotype-targeted therapy is the fact that the circulating M-protein acts as a sink for the targeting moiety during the active phases of MM disease. At these stages, however, conventional drugs are often able to dampen MM to the stage of minimal residual disease (MRD), when patients have achieved a deep response, but residual MM cells may remain. These residual cells must be targeted with the highest possible precision in order to achieve the complete eradication of all residual and dormant tumor cells and thereby break the pattern of patient relapse. In this regard, the idiotype represents an ideal drug target, as it is a cancer cell-exclusive antigen. The use of the BCR idiotype as a cancer target is, however, complicated by the fact that it is patient-specific and, therefore, not universally applicable. Despite these two major difficulties to overcome in idiotype-targeted MM treatment, some (clinical) efforts to exploit this antigen have already been made, mainly in cancer vaccination strategies [35,36,37,38,39,40,41,42,43,44]. These efforts highlight the clinical interest in further exploring the MM idiotype as a drug target. 

Our laboratory has previously developed and described anti-idiotypic VHHs, which were developed as tracer molecules for in vivo biomolecular imaging and (combined) targeted radionuclide therapy for MM. This was performed against the idiotype of both the murine 5T2 (5T2MMId) and 5T33 (5T33MMId) MM models [35,45,46]. These MM models were chosen as they are fully immunocompetent, and pathogenesis, myeloma progression, and cancer cell localization strongly resemble human disease, rendering them highly relevant to the study of the course of disease and treatment of MM [47,48]. Additionally, the feasibility of generating VHHs that specifically recognize the idiotype of human patient-derived MM cells was also demonstrated [35].

In the current study, we evaluated the potential of using 5T33MMId-specific VHHs as targeting domains in VHH-based CAR-T cells. With this, we provide a proof-of-concept for a strategy to develop personalized therapies for MM patients, using VHHs that offer multiple advantages for precision medicine, including the non-invasive detection of MM cells and the ability to generate CAR-T cell therapies to tackle residual MM cells. 

## 2. Results

### 2.1. High-Affinity VHHs Are the Most Capable In Vivo Tracer Moieties

In previous studies, we generated 5T2MMId- and 5T33MMId-specific VHHs [35,45]. The molecular characterization of the 5T33MMId-specific VHHs included, amongst others, affinity parameter determination (k_a_, k_d_, and K_D_) via surface plasmon resonance (SPR) towards the purified idiotype protein, and the evaluation of their in vivo biodistribution. To that end, the VHHs were radiolabeled and injected intravenously into both healthy and 5T33MM-bearing animals. An increase in radioactive signals in MM-colonized organs or tissues in diseased mice compared to healthy animals, was considered a measure of specific VHH uptake. The observed in vitro and in vivo characteristics are shown in Table 1, in order of decreasing specific in vivo VHH uptake. The 5T2MMId-specific VHH-R3B23 was included as a negative control, in both the previous and this study, as it is specific to a different murine MM idiotype protein, confirming idiotype specificity [35,45].

We further determined the binding capacities of these VHHs against the membrane-anchored antigen with flow cytometry and observed strong binding capacities (K_D_ < 10 nM) for the majority of the compounds (Figure 1). 

The observed K_D_ values were in line with the mediocre to high specific VHH uptake observed in previous animal experiments (Table 1). An exception was VHH-8311, which appeared to have a high affinity toward the purified 5T33MMId protein and the membrane-anchored 5T33MMId antigen in both prior and our data, but for which the specific in vivo uptake was lower. However, this was mainly due to the elevated aspecific retention of the compound in healthy organs, which lowered the specific uptake in tumor lesions. VHHs 8404 and 8335 displayed higher K_D_ values (i.e., weaker affinities for the membrane-anchored idiotype), and whilst the in vivo uptake of VHH-8335 could not be determined, VHH-8404 showed lower specific accumulation. This leads to the general conclusion that there is a fairly positive relationship between the affinity of a compound for its membrane-anchored target and its capacity as an in vivo tracer molecule. 

### 2.2. Reporter T Cells Can Be Stably Transduced with Different Correctly Folded Anti-Idiotypic VHH-CARs

The VHHs were, next, cloned into a murine second-generation 4-1BB co-stimulatory CAR backbone in a lentiviral transfer plasmid, as schematically displayed in Figure 2. 

The different VHH-CAR-containing lentiviral transfer plasmids were, next, co-transfected with packaging and envelope plasmids into HEK293T lentiviral particle (LV)-producing cells. In this way, different batches of LVs were generated, capable of stably transducing cells with different VHH-CARs. These CARs were identical, with the exception of the antigen-binding portion, i.e., the VHH. By transducing the murine DO11.10 reporter T cell line, several CAR-T cell lines were obtained. DO11.10 cells act as T cell activation reporter cells via the production of (murine) interleukin-2 (IL-2). Figure 3 displays the successful DO11.10 cell transduction with the different 5T33MMId-specific VHH-CARs and a non-targeting control VHH-CAR that incorporated the 5T2MMId-specific VHH-R3B23.

The transduction rates of the 5T33MMId-specific CARs were similar (Figure 3), allowing a side-by-side comparison of the T cell activation potential. Since the flow cytometry staining was performed with a soluble antigen for the 5T33MMId-specific CARs and with an anti-VHH conformation mAb for the control CAR, not only was the success of transduction demonstrated but also the correct expression and protein folding of the CARs on the surfaces of the DO11.10 T cells.

### 2.3. VHH-Based Anti-Idiotypic CARs Can Induce T Cell Signaling in Reporter T Cells, but the Majority of VHH-CARs Fail

To evaluate the differences in CAR-T cell activation kinetics, co-culture experiments were performed by co-incubating VHH-CAR-expressing DO11.10 T cells without (unstimulated condition) or with (stimulated condition) 5T33vt MM target cells at a 1:1 CAR-T cell–target cell ratio and measuring IL-2 concentrations in the culture supernatants at different time-points. Specific T cell activation was defined as the difference in the amount of IL-2 produced between the stimulated and unstimulated conditions (Δ_[IL-2]_). Surprisingly, not all CARs induced T cell activation, and large differences were observed within the evaluated time period. In addition to the non-transduced (wild-type (wt)) DO11.10 cell line, CARs that included the negative control VHH-R3B23 and 5T33MMId-specific VHHs 8335, 8351, 8379, and 8404 also did not lead to any expression of IL-2 (Figure 4A). VHH-8311 provided a mild upregulation of IL-2, while VHHs 8326 and 8387 were the only ones to enable strong IL-2 production, i.e., T cell activation. These experiments allowed us to identify a time-point at 40 h post-co-incubation for a more full-fledged side-by-side comparison of CAR-T cell activation as, at that moment, a plateau in IL-2 production was noted.

The side-by-side CAR-T cell activation assays confirmed previous findings, namely, that only VHHs 8311 (Δ_[IL-2]_ = 238.01 ± 9.10 pg/mL), 8326 (Δ_[IL-2]_ = 322.52 ± 23.95 pg/mL), and 8387 (Δ_[IL-2]_ = 346.64 ± 7.01 pg/mL) are CAR-T cell-activating VHHs. These give rise to a significant upregulation of IL-2 production compared to the non-targeting control VHH-R3B23 (Δ_[IL-2]_ = 0.06 ± 1.25 pg/mL; *p*-values < 0.0001). Further confirming earlier results, none of the other VHH-CARs evoked T-cell activation, with Δ_[IL-2]_ = 1.14 ± 1.48 pg/mL for VHH-8335 (*p*-value = 0.9998), Δ_[IL-2]_ = 5.32 ± 2.41 pg/mL for VHH-8351 (*p*-value = 0.9756), Δ_[IL-2]_ = 2.77 ± 0.54 pg/mL for VHH-8379 (*p*-value = 0.9994), and Δ_[IL-2]_ = 6.26 ± 4.24 pg/mL for VHH-8404 (*p*-value = 0.9435), despite their specificity and high affinity for the target antigen (Figure 4B and Table 1) [35].

### 2.4. The VHH-Targeted Epitope Is Non-Determinant for CAR-T Cell Activation

As the affinities of the VHHs for 5T33MMId were not determinant of the degree of CAR-T cell activation, we next investigated whether the specific VHH-bound epitope is a controlling factor. This is directly related to the intercellular distance between the T cell and target cell and has, therefore, been opted as an important parameter for adequate T cell activation. To verify this, we performed a series of co-culture T cell activation experiments, as described before, this time preincubating the target cells with a saturating concentration of soluble VHH. As such, the binding of a soluble VHH to an epitope similar to the one bound by the CAR-incorporated VHH causes the blocking of CAR-T cell activation, measured as a lowering of IL-2 production. Thus, if a lowering of IL-2 production is observed, the VHHs bind to an overlapping epitope. Since only VHHs 8311, 8326, and 8387 gave rise to sufficient levels of IL-2 (i.e., CAR-T cell activation), only these VHH-CAR cell lines were evaluated in competition assays. These studies showed that all VHHs, including the ones incapable of activating the CAR-T cells, interfere with and negatively affect each other’s binding to the 5T33MMId antigen, implying overlapping binding sites on the target protein (Figure 5). 

Given the very small and specific nature of the antigen targeted by the VHHs (i.e., the idiotype, which is the three-dimensional protein structure created by the six hypervariable CDR regions combined), this general competition is not surprising from a structural point of view. However, it is notable that even though all VHHs bind to the antigen at a structurally similar location, major differences in the capacity of CAR-T cell activation were observed (Figure 4B). This shows that in addition to the affinity of a VHH for its target, the targeted epitope and the related intercellular distance between the T cell and the target cell also have no direct predictive value for the degree of T cell activation. 

These results highlight the unpredictability of CAR functioning, depending on the antigen-recognizing domain that is incorporated. This also shows that the importance of the careful selection of a VHH (or another antigen-binding moiety) directly in the desired context (i.e., in a CAR format) should not be underestimated and that it is crucial to carefully select the antigen-binding domain on a case-by-case basis.

## 3. Discussion

In the present study, we are the first to report on the successful generation of idiotype-specific VHH-based CAR-T cells in the context of MM. We performed a side-by-side evaluation of multiple idiotype-specific VHH-CARs that only differ in their antigen-binding domain (i.e., the incorporated VHH). Important differences in their ability to mediate IL-2 secretion, a critical cytokine affecting T cell differentiation and activation, were observed in reporter T cell lines, despite the fact that (i) all VHHs exhibited a strong affinity for the antigen, which was also confirmed toward the membrane-anchored protein; (ii) they were all capable of specifically targeting MM cells in vivo to different extents, largely in line with their observed affinities; and (iii) they all bound to similar epitopes on the target protein, as observed in the competition set-up of the CAR-T cell activation assays. Therefore, our results highlight the importance of the side-by-side evaluation of multiple antigen-binding moieties in a CAR and careful selection thereof.

Currently approved BCMA-specific CAR-T cell products initially achieve impressive remission rates in heavily pre-treated MM patients [10,49], but relapse is still commonly observed [11,12]. Two main reasons for this are the use of BCMA as a CAR-T cell target, on the one hand [11,12], and the artificially formatted antigen-recognizing domains on the other [11,18,50], which will be further addressed in the following sections. 

Selecting an appropriate and robust antigen is key to the success of CAR-T cell therapy or targeted therapy in general. The currently approved products Abecma^®^ and Carvykti^®^ both target BCMA. Theoretically, BCMA is a favorable MM target antigen, as it is expressed almost exclusively by late-stage B cells and is important for their survival [11,51]. However, BCMA loss or downregulation, be it transient or not, has been observed upon patient relapse [11,12]. This may be due to enhanced γ-secretase-mediated shedding [10,20,21] or genomic loss [22,27]. Therefore, many other MM targets are under evaluation, as reviewed by Mishra et al. [52]. An ideal target is present exclusively on cancer cells and is resistant to shedding or other forms of downregulation. Any expression of the antigen on healthy cells is accompanied by a certain degree of toxicity [53]. Although this form of on-target-off-tumor toxicity is largely considered manageable for hematological cancers, it is typically associated with impaired immune functioning, implying the need for prolonged treatment [54]. In this sense, the idiotype of MM cells represents a perfect drug target at a point in time when the levels of the circulating paraprotein are low, a stage that is commonly achieved after initial lines of conventional therapy [2,8,55,56,57,58]. Notoriously, at this stage, however, most patients are still in a state of MRD, in which a small but dangerous population of dormant MM cells is present that may act as a source of relapse [59]. It is established that MM cells express a membrane-anchored idiotype in a large portion of patients [31,32,33,35] and that this expression remains present in these dormant cells [34]. The idiotype of MM is a cancer cell-exclusive antigen, which is not the case for the other MM antigens under investigation, and, therefore, is an ideal marker for these dormant cells, that need to be detected and eliminated with the highest precision. Yet, the gain in cancer cell exclusivity comes with patient specificity and a loss of universal applicability. This renders idiotype-targeted treatment more complex compared to other targets. 

A second major limitation of currently approved CAR-T cells for MM is CAR-format-related. Abecma^®^ and Carvykti^®^ are identical from the hinge down, as they both harbor CD8α-derived hinges and transmembrane domains, and 4-1BB co-stimulation and CD3ζ T cell activation units [49]. In terms of antigen recognition, Abecma^®^ is more classical compared to Carvykti^®^, as it incorporates a mAb-derived scFv against BCMA [49].

ScFvs are typically incorporated into CARs to achieve antigen recognition. They consist of the artificially linked V_H_ and V_L_ domains of a mAb, which, in natural conditions, are held together by hydrophobic interactions and further sustained by cysteine bonds between neighboring protein domains. By removing the latter, the scFv largely loses its integrity, and it becomes necessary to insert a (potentially immunogenic) linker between the V_H_ and V_L_ domains, hence the term ‘single-chain’ variable fragment. Furthermore, the increased freedom of movement of the V_H_ and V_L_ domains relative to each other allows the hydrophobic protein patches to potentially mismatch and/or aggregate [60]. The domain swapping and mismatching of the V_H_ and V_L_ domains can result in receptor clustering, which is associated with tonic (antigen-independent) signaling and, consequently, premature T cell exhaustion [14]. These intrinsic stability issues are the first reason why scFvs are not optimal as part of the receptor [16]. 

In addition, clinical mAbs, from which these scFvs are usually directly derived, often have non-human (mainly murine) origins and, therefore, may be immunogenic. Indeed, it has been shown that many cases of relapse after CAR-T cell therapy are due to anti-CAR immune responses with a therapy-neutralizing effect, allowing MM cell regrowth [10,11,13,61]. VHHs are relatively small and show strong sequence similarities to human V_H_-III family domains [62]. Therefore, they are considered less immunogenic in comparison with scFvs. In Carvykti^®^, antigen recognition is provided by two fused bi-epitopic, BCMA-specific VHH molecules [49]. This means that the size and build of the targeting moiety do not differ substantially from those of an scFv, but the origin and mode of targeting do. Due to this dual BCMA targeting, the gain in avidity will compensate for the potential loss in affinity that may arise from a reduction in the total size of the paratope from six CDRs to three [63]. However, by doing so, the advantages of the VHHs’ smaller size and monomeric nature are lost, as an artificial linker is needed for the VHH fusion. These two factors, again, increase the CARs’ immunogenicity. Hence, single-domain CARs in which only one VHH is used for antigen recognition are theoretically preferred over this artificial format. Of note is that concerns about the non-human origin of VHHs remain. However, humanization protocols are available [24], which, in principle, further reduce the risk of a host reaction against the CAR. Alternatively, efforts are being made to use human V_H_ domains, in which the hydrophobic protein stretches that are responsible for V_L_ association are ‘camelized’ to obtain VHH-like solubility [13,24,64].

In addition to the suggested advantage in terms of immunogenicity, VHHs have another advantage over scFvs, namely, that they are selected from relatively small immune libraries [65]. In an scFv, a correctly paired, i.e., simultaneously matured, V_H_ and V_L_ combination is required to achieve full-fledged antigen recognition. However, these are not directly joined on a genomic level, and the pairing is, therefore, lost during in vitro selection procedures, resulting in both a V_H_ and a V_L_ immune library, both of a similar size to a VHH library. The complete immune scFv library consists of all possible combinations of these V_H_ and V_L_ molecules and is, therefore, enormous compared to a VHH library. This is one of the major reasons why scFvs are usually not selected from such libraries but are, rather, adapted from already known and (clinically) validated mAbs. This, in turn, implies that they are not optimized to function in a CAR. Our data directly demonstrate that this approach does not lead to the most optimal compounds for T cell activation, as it has been shown that some high-affinity VHHs do not lead to T cell activation, despite many favorable in vitro and in vivo characteristics in soluble form. 

Based on previous data, VHH-8379 was selected as the lead compound for further development as a radionuclide tracer, as it was considered optimal for in vivo molecular imaging and targeted radionuclide therapy in a physiologically relevant MM mouse model [35]. The main selection criteria for this were high specific tracer accumulation in MM tumors and tolerable kidney retention, which is the dose-limiting factor in radioactive diagnostic or therapeutic applications [66]. However, our data show that this VHH is not competent when incorporated into a CAR, as it does not induce any T cell activation. 

The fact that different VHHs are needed when a different application is envisaged requires that different VHHs are selected. When the BCR idiotype is chosen as a drug target, tracer selection is already complicated by the fact that the antigen is highly individual and that the VHHs must be designed patient-specifically, implying that a different VHH selection process must be completed for each patient. Furthermore, our data show that not only should a VHH selection process should be carried out, but also a VHH-CAR selection procedure. The relatively simple basis for VHH screening (i.e., relatively small immune libraries) makes it more suited to achieving this compared to very large and complex or artificial scFv libraries. Nonetheless, the requirement of patient-specific VHH generation within an acceptable timeframe remains an important challenge ahead. On the bright side, the fact that different VHHs act optimally as tracers or as CAR components could also be an advantage, as they can then potentially be used synergistically during disease treatment and follow-up. In the clinic, increasing attention is being paid to personalized medicine, where diagnostics are patient-specific and confluent with therapy [67]. Having a VHH that monitors a patient’s disease and antigen expression status prior to and during treatment could be of tremendous added value. This would be possible, for example, with targeted radionuclide imaging, an application for which VHHs are ideally suited [68], in combination with CAR-T cell treatment. 

## 4. Conclusions

All in all, in this study, we demonstrated an important shortcoming of the common approach of CAR design, in which the antigen-binding moiety is simply adopted from existing and clinically validated (murine) mAbs. Particularly, the mere incorporation of a targeting moiety that was originally developed for a different purpose does not necessarily lead to (optimally) functioning CARs. The importance of the attentive case-by-case selection of the antigen-binding moiety has only recently been gaining more attention. However, our data underline the relevance of this. Finally, we provided a proof-of-concept for anti-idiotype CAR-T cell therapy, which represents a personalized therapy for MM patients that would be highly valuable to tackle residual MM cells and, as such, prevent disease relapse.

## 5. Materials and Methods

### 5.1. Cell Culture 

Bacterial *E. coli* NEB5-α (New England BioLabs, Ipswich, MA, USA) and *E. coli* XL1-Blue (Agilent, Zaventem, Belgium) cultures were grown in lysogeny broth medium and used for initial and large-scale plasmid production, respectively. The murine 5T33vt MM cell line was previously described in [35] and was cultured in a supplemented Roswell Park Memorial Institute 1640 culture medium (Gibco^TM^, Thermo Fisher Scientific, Waltham, MA, USA). Human embryonic kidney (HEK) 293T cells were purchased from the American Type Culture Collection (ATCC, Manassas, VA, USA) and used for the production of LVs. The H-2-I-Ad-restricted OVA-specific murine T cell hybridoma DO11.10 recognizes an I-Ad-binding epitope located within amino acids 323–339 of the OVA protein sequence [69]. This cell line was provided by Dr. P. Marrack (the National Jewish Health Center, Denver, CO, USA) and served as an IL-2 reporter cell line for T cell activation. Both cell lines were cultivated in supplemented Dulbecco’s modified Eagle’s medium (Gibco^TM^). ‘Supplemented’ refers to the addition of 10% fetal bovine serum (Thermo Fisher Scientific, Waltham, MA, USA), 2 mM L-Glutamine (Thermo Fisher Scientific), 1 mM sodium pyruvate (Thermo Fisher Scientific), 1% penicillin–streptomycin (Thermo Fisher Scientific), and 1% non-essential amino acids (Thermo Fisher Scientific). 

### 5.2. VHHs

The 5T2MMId- and 5T33MMId-specific VHHs used in this study were previously described by Lemaire et al. [45], and Puttemans et al. [35], respectively. The cloning into the pHEN6c production vector [70], transformation of the *E.coli* WK6 VHH production strain, induction of periplasmic protein production and subsequent protein purification via osmotic shock, immobilized metal affinity chromatography and size exclusion chromatography to yield C-terminally hexahistidine-tagged VHHs, were performed as previously described in [71].

### 5.3. Assessment of VHH Binding to Cells and Affinity Determination

For the determination of affinity toward the membrane-anchored protein, 5T33vt cells were incubated with the hexahistidine-tagged VHHs in a 1/3 serial dilution, ranging from a 0 nM to a 333 nM final VHH concentration, for 1 h at 4 °C. The staining steps included a primary murine anti-hexahistidine IgG1 mAb (Biolegend, San Diego, CA, USA) and a secondary APC-labeled anti-mouse IgG1 mAb (Biolegend). Antigen expression by 5T33vt cells was confirmed by subsequent staining with an anti-5T33MMId IgG1 mAb [48] and a PE-labeled anti-IgG1 mAb (BD Biosciences, Franklin Lakes, NJ, USA). The staining procedures were executed according to the manufacturer’s instructions. Cell fluorescence was measured with the BD Biosciences FACSCelesta apparatus, and data analysis was carried out with the FlowJo 10.9.0 software (BD Biosciences). For the affinity determination, the data were normalized to 100% for the highest value measured and to 0% for the data point where no VHHs were added, after background subtraction. Curve fitting to determine the K_D_ values was performed using a non-linear fit [agonist] vs. normalized response model with the GraphPad Prism 9.1.0. software. 

### 5.4. Design and Cloning of a Murine Lentiviral VHH-CAR Transfer Plasmid

A previously described pHR’-derived lentiviral transfer plasmid was used for the generation of LVs that are capable of stably transducing target T cells with a desired genetic construct [72]. In this case, the genetic construct encoded a fully murine, second-generation CAR sequence, as depicted in Figure 2. The designed CAR encodes an EF1α core promotor and Igκ leader sequence, followed by a VHH-cloning site, a murine CD8α-derived hinge and transmembrane region, a murine 4-1BB co-stimulatory unit, and a murine CD3ζ-derived T cell activation domain (Figure 2). The complete DNA sequence of the CAR construct used is provided in Appendix A.

This DNA molecule, flanked by 20-nucleotide-long pHR’ overhangs, was ordered to demand from Integrated DNA Technologies (IDT) (Figure 2, panel below). The commercialized Gibson assembly method (IDT, Coralville, IA, USA) was executed to integrate the ordered CAR sequence into the triple helix and 3′ ∆U3 long terminal repeat-containing pHR’ backbone, after ClaI/XhoI vector linearization (Thermo Fisher Scientific). PstI/BstEII restriction (Thermo Fisher Scientific) followed by T4 DNA ligation (Thermo Fisher Scientific) ensured the VHH’s incorporation into the VHH-CAR-containing lentiviral backbone. The transformation of competent NEB5-α *E. coli* (New England Biolabs), firstly, and XL1-Blue *E. coli* (TransformAid Bacterial Transformation Kit; Thermo Fisher Scientific), secondly, allowed for large-scale plasmid production (NucleoBond Xtra Maxi kit; Macherey-Nagel, Düren, Germany). OD_260_/OD_280_ measurements were performed for the assessment of the yield and quality of the obtained plasmids. All plasmids were fully sequenced for quality control (NightXpress Mix2Seq Kit, Eurofins Genomics, Konstanz, Germany).

### 5.5. LV Production and T Cell Transduction

The production of LVs encoding the different VHH-CARs was performed via the co-transfection of HEK293T cells with lentiviral transfer plasmids (the VHH-CAR pHR’ plasmids described above), packaging plasmids (pCMVΔR8.9), and envelope-encoding plasmids (pMD.G) at a 3:2:1 plasmid ratio via procedures previously described in [72,73]. The packaging and envelope plasmids were kindly gifted by Prof. Trono (University of Geneva Medical School, Geneva, Switzerland). The HEK293T culture supernatant containing the produced LVs was collected at 48 and 72 h post-transfection, supplemented with 10 µg/mL of protamine sulfate (LeoPharma, Ballerup, Denmark), and directly used for the transduction of reporter DO11.10 T cells. 

### 5.6. Evaluation of Transduction Efficiency

Transduction rates were measured via flow cytometry by subsequent staining with 100 nM of the 5T33 IgG2b mAb [48] and the FITC-labeled anti-mouse IgG2b mAb (BD Biosciences). The VHH-R3B23-CAR was subsequently stained with a biotinylated anti-VHH mAb (MonoRab Rabbit Anti-Camelid VHH mAb, GenScript, Rijswijk, The Netherlands) and streptavidin–PE (BD Biosciences). All staining steps were performed according to the manufacturer’s instructions. The cell fluorescence was measured with a BD Biosciences FACSCelesta flow cytometer, and the data analysis and visualization were performed with the FlowJo 10.9.0 software (BD Biosciences). 

### 5.7. Assessment of T Cell Activation by Different VHH-CARs

Different VHH-CAR-expressing DO11.10 cell lines were co-incubated at a 1:1 ratio with or without 5T33MMId^pos^ 5T33vt target cells (referred to as the stimulated and unstimulated conditions, respectively) in the DO11.10 culture medium at 37 °C. Murine IL-2 produced in the co-culture supernatant was measured three times a day over a period of 96 h via an IL-2 mouse ELISA Kit (Invitrogen, Carlsbad, CA, USA) in order to identify a valid time-point for the comparative assessment of T cell activation. This was pinpointed at 40 h post-co-incubation. Specific activation was defined as the difference in IL-2 produced between the stimulated and unstimulated conditions. In the competition set-up, 5T33vt cells were pre-incubated with a 500 mM saturating concentration of soluble VHH for 1 h at 37 °C prior to co-incubation with the reporter (CAR-)T cells at a 1:1 cell ratio. Residual specific T cell activation was assessed at 40 h post-co-incubation via IL-2 ELISA measurements (Invitrogen). 

### 5.8. Statistical Analysis

For the statistical analysis of the VHH-CAR-T cell activation assays, the specific CAR-T cell activation data were assumed to have a normal distribution. Statistical analysis was performed by one-way ANOVA (multiple *t*-tests), where each 5T33MMId-specific DO11.10-CAR cell line was compared to the non-targeting control DO11.10-CAR-R3B23 cell line. Analyses were performed using GraphPad Prism 9.1.0. (* *p* < 0.05; ** *p* < 0.01; *** *p* < 0.001; **** *p* < 0.0001; not significant (n.s.), *p* > 0.05). 

## Figures and Tables

**Figure 1 ijms-25-05634-f001:**
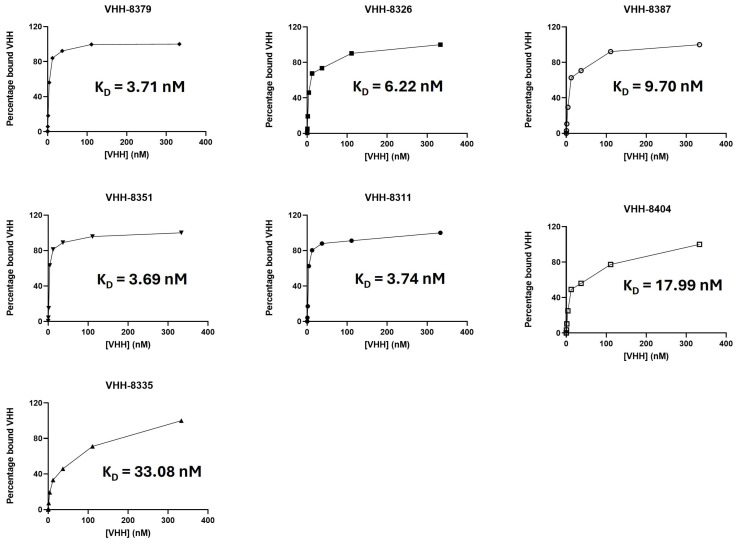
**Determination of VHH affinity toward membrane-anchored 5T33MMId antigen with flow cytometry.** A 1/3 VHH dilution series was incubated with 5T33MMId^pos^ 5T33vt cells. Cell binding was assessed via flow cytometry after staining for the VHH-hexahistidine tag. Affinity parameter calculation and data visualization were performed using GraphPad Prism 9.1.0. (*n* = 1).

**Figure 2 ijms-25-05634-f002:**
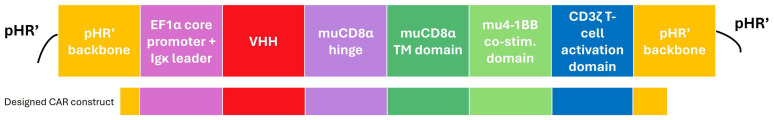
**Overview of the used CAR construct in the pHR’ lentiviral transfer backbone plasmid.** This second-generation 4-1BB-based murine CAR construct includes an EF1α core promotor and Igκ leader sequence, a VHH-cloning site, CD8α-derived hinge and transmembrane regions, and a 4-1BB co-stimulatory and CD3ζ T cell activation domain. EF1α = eukaryotic translation elongation factor 1α; TM = transmembrane; mu = murine; co-stim. = co-stimulatory.

**Figure 3 ijms-25-05634-f003:**
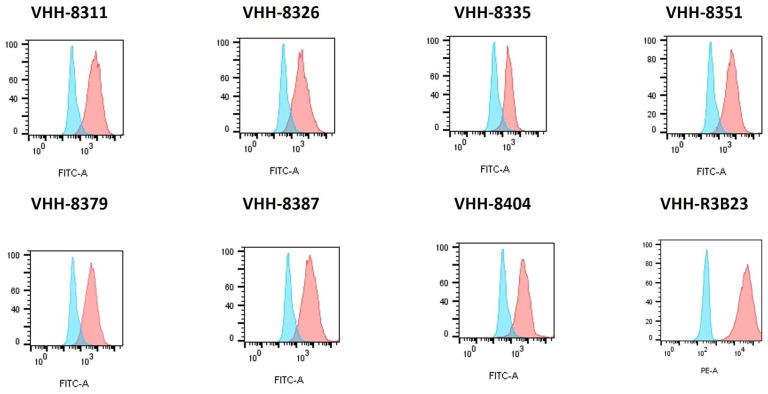
**Assessment of T cell transduction rates of the different VHH-CAR-DO11.10 cell lines via flow cytometry.** Transduction rates and functional CAR expression on the surfaces of reporter DO11.10 cell lines transduced with the different 5T33MMId-specific VHH-CARs and the non-targeting negative control VHH-CAR R3B23 were assessed via flow cytometry. Cell staining was performed via the addition and detection of the soluble 5T33MMid antigen for the idiotype-specific cell lines and with anti-VHH mAb for the negative control cell line D011.10-VHH-R3B23. Red histograms show VHH-CAR transduced D011.10 cells and are displayed relative to equally stained untransduced DO11.10 cells (blue histograms) (*n* = 1).

**Figure 4 ijms-25-05634-f004:**
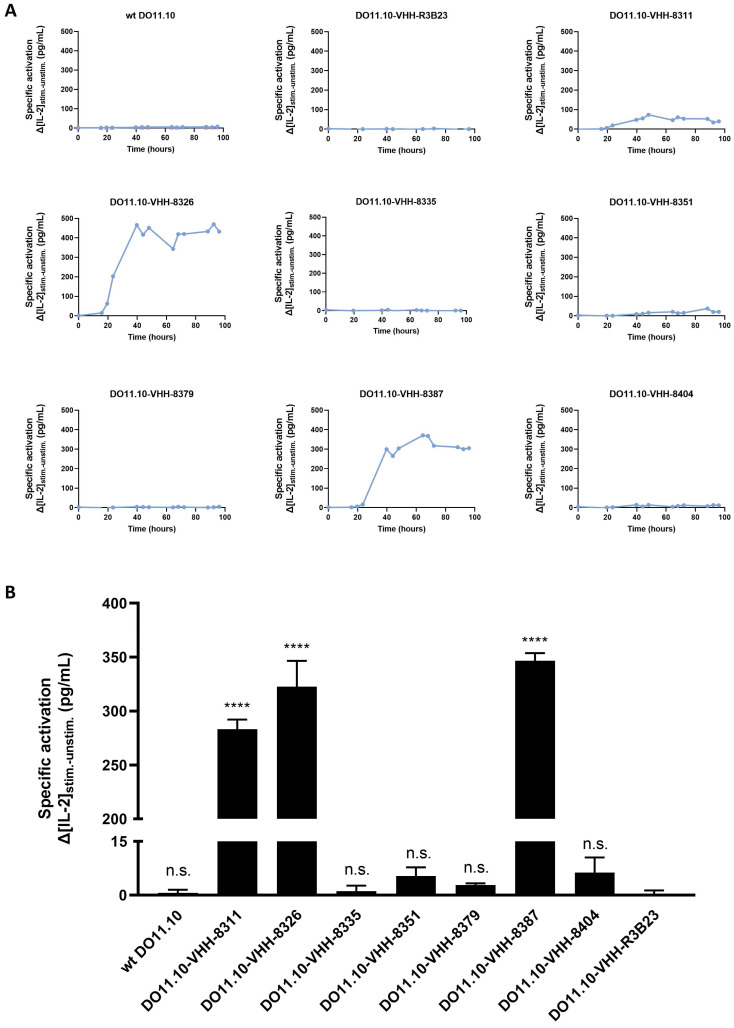
**Specific CAR-T cell activation by the different VHH-CAR-transduced DO11.10 reporter T cell lines.** (**A**): Differences in IL-2 production between (1:1) co-cultured DO11.10-VHH-CAR-T cells with 5T33MMId^pos^ 5T33vt cells (stimulated condition) and without stimulator cells (unstimulated condition), as determined via IL-2 ELISA over a period of 96 h (*n* = 1). (**B**): Differences in IL-2 production between 40 h (1:1) co-cultured DO11.10-VHH-CAR-T cells with 5T33MMId^pos^ 5T33vt cells (stimulated condition) and without stimulator cells (unstimulated condition), as determined via IL-2 ELISA. Graphs represent the mean differences in IL-2 production ± SD (*n* = 3). **** *p* < 0.0001; not significant (n.s.), *p* > 0.05.

**Figure 5 ijms-25-05634-f005:**
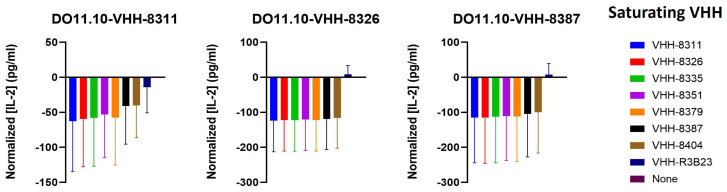
**Assessment of T cell activation by different VHH-CARs in a competition set-up to evaluate the VHH-bound epitope.** Graphs show the IL-2 production rates of 40 h 1:1 co-cultures of VHH-CAR-transduced DO11.10 cell lines with 5T33MMId^pos^ 5T33vt cells in the presence of 1 µM excess of soluble VHH to determine binding competition for a similar epitope on the idiotype antigen. Data are quantified in terms of mean concentrations of IL-2 produced ± SD after normalizing to the condition in which no soluble VHH is added to the co-culture (*n* = 3).

**Table 1 ijms-25-05634-t001:** Previously determined in vitro and in vivo characteristics of the purified 5T33MMId-specific VHHs. Adapted with permission from Puttemans et al. [35], 2021, Nick Devoogdt. NA = not analyzed.

VHH	*k*_a_ (M^−1^s^−1^)	*k*_d_ (s^−1^)	*K*_D_(M)	Specific In Vivo Uptake
**8379 ***	2.83 × 10^5^	5.48 × 10^−4^	1.94 × 10^−9^	High
**8326**	9.84 × 10^5^	10.6 × 10^−4^	1.08 × 10^−9^	High
**8387**	3.18 × 10^5^	12.8 × 10^−4^	4.03 × 10^−9^	Mediocre
**8351**	4.66 × 10^5^	5.93 × 10^−4^	1.27 × 10^−9^	Mediocre
**8311**	6.27 × 10^5^	2.39 × 10^−9^	0.38 × 10^−9^	Low
**8404**	2.67 × 10^5^	14.6 × 10^−4^	5.47 × 10^−9^	Low
**8335**	10.6 × 10^5^	23.1 × 10^−4^	2.18 × 10^−9^	NA ****
**R3B23**	Not binding	Not binding	Not binding	None

* Due to its favorable characteristics, VHH-8379 was selected as a lead candidate for targeted radionuclide therapy. ** VHH-8335 could not be radiolabeled with sufficient purity and was, therefore, excluded from in vivo biodistribution studies.

## Data Availability

The original contributions presented in this study are included in this article/the Appendix A. Further inquiries can be directed to the corresponding author/s.

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
