# Peer review of "Anti-Idiotypic VHHs and VHH-CAR-T Cells to Tackle Multiple Myeloma: Different Applications Call for Different Antigen-Binding Moieties"

_ijms, 2024, doi:10.3390/ijms25115634_

Round 1

Reviewer 1 Report

Comments and Suggestions for Authors

The manuscript by Heleen Hanssens and co-workers is titled “Anti-idiotypic VHHs and VHH-CAR-T cells to tackle multiple myeloma: different applications call for different antigen-binding moieties.” It is well-written and easy to read. It adds clustering information on an extremely interesting topic. Multiple Myeloma is still incurable the success of CAR T approaches is limited in time. Unfortunately, patients' relapse is still frequently reported, and today, there isn't a valid and shared protocol for CAR T therapy. 

Minor: The introduction should be more focused. Part of it should moved to discussion. 

Maior: The CAR construct sequences must be described in the supplementary file, which at the moment is the word of the full manuscript. In particular, the sequence of the CD3ζ T-cell activation domain. 

Reviewer 2 Report

Comments and Suggestions for Authors

Reviewer comments and suggestions

The authors in this study characterized VHHs that specifically target the idiotype of murine 5T33 multiple myeloma (MM) cells. The idiotype is a highly cancer-specific but also patient-specific target antigen, making it one of the most promising yet difficult MM targets. These were incorporated into variable heavy domains of heavy chain (VHH)-based CAR modules, of which the format has advantages compared to scFv-based CARs. It has been reported that VHHs previously selected as lead compounds for targeted MM radiotherapy do not correspond to the best (CAR) T-cell activators. As a result, the study emphasizes the significance of choosing a particular VHH based on its intended use, highlighting a significant flaw in the prevalent CAR development methodologies used today.

Overall, the manuscript was good. However, a few major concerns/comments needed to be explained or modified. 

  1. Line 18 BCMA first time used should be in full form.
  2. Line 66 chimeric antigen receptors you can add near the abbreviations.
  3. Line 76-77 Please explain it in a better way
  4. Line 130-133 is there any validation for the study in the animal model of MM
  5. Line 154-155 Not everything should be present in the introduction section, please delete these lines
  6. Line 319-321 you can explain the reason either in the introduction or discussion section completely.
  7. Line 375-381 it would be nice if they mentioned the table or figure number at the respective place.
  8. Line 502-505 The authors suggested invivo but could not find any information regarding this in the material and method paragraph.
  9. References should be based on MDPI guidelines, which all need to be modified.

Reviewer 3 Report

Comments and Suggestions for Authors

Dear authors,

The work that you present in this manuscript focuses on the treatment of multiple myeloma, considering the CAR-T cell therapy. The aim of the study was to evaluate the potential of specific VHHs as targeting domains of VHH-based CAR-T cells. A personalized therapy for patients with multiple myeloma is of great interest dur to its multiple advantages.

The manuscript is well written, the references are well chosen and in agreement with the subject.

I suggest the reduction of table 1 and the enlargement of figures 1 and 4A.

Even if the Discussion part is well detailed, I consider that a Conclusion part could make the manuscript easier to read and understand. 
